# An intravital window to image the colon in real time

Nikolai Rakhilin[1,2,7], Aliesha Garrett [1,7], Chi-Yong Eom[3], Katherine Ramos Chavez[1], David M. Small[3], Andrea R. Daniel[4], Melanie M. Kaelberer[5], Menansili A. Mejooli[3], Qiang Huang[1], Shengli Ding[1], David G. Kirsch [4,6], Diego V. Bohórquez [5], Nozomi Nishimura [3], Bradley B. Barth [1]* & Xiling Shen[1]*

Intravital microscopy is a powerful technique to observe dynamic processes with single-cell resolution in live animals. No intravital window has been developed for imaging the colon due to its anatomic location and motility, although the colon is a key organ where the majority of microbiota reside and common diseases such as inflammatory bowel disease, functional gastrointestinal disorders, and colon cancer occur. Here we describe an intravital murine colonic window with a stabilizing ferromagnetic scaffold for chronic imaging, minimizing motion artifacts while maximizing long-term survival by preventing colonic obstruction. Using this setup, we image fluorescently-labeled stem cells, bacteria, and immune cells in live animal colons. Furthermore, we image nerve activity via calcium imaging in real time to demonstrate that electrical sacral nerve stimulation can activate colonic enteric neurons. The simple implantable apparatus enables visualization of live processes in the colon, which will open the window to a broad range of studies.

[1] Department of Biomedical Engineering, Duke University, Durham, NC 27710, USA. [2] School of Electrical and Computer Engineering, Cornell University, Ithaca, NY 14853, USA. [3] School of Biomedical Engineering, Cornell University, Ithaca, NY 14853, USA. [4] Department of Radiation Oncology, Duke University Medical Center, Durham, NC 27710, USA. [5] Department of Medicine, Duke University Medical Center, Durham, NC 27710, USA. [6] Department of Pharmacology & Cancer Biology, Duke University Medical Center, Durham, NC 27710, USA. [7] These authors contributed equally: Nikolai Rakhilin, Aliesha Garrett. *email: bradley.barth@duke.edu; xiling.shen@duke.edu

Intravital microscopy is a powerful technique for observing various cellular processes in live animals[1]. Techniques for intravital microscopy via surgically implanted windows have been developed for many organs including the brain[2–4], spinal cord[5], liver[6,7], lung[8], skin[9,10], and small intestine[11–13], among others[14]. These windows can be maintained in mice for weeks at a time, allowing high resolution, chronic fluorescent imaging of the organs and tissues[14]. However, no such techniques have been developed for the colon to date, which is a particularly challenging target due to the high amplitude and frequency of distension and sensitivity to obstructions. There is a need for a colonic window to visualize the dynamics of the colon, such as the epithelial stem cell population, the immune response to inflammatory injury, and the neural circuitry the mediate colonic function. This new technology emphasizes chronic intravital imaging, making it possible to track the same location in colon in the same subject for multiple days.

The colon is part of the lower gastrointestinal tract and plays a key role in the absorption of water, salt, and other nutrients from digested chyme. Over half of the body's gut flora reside in the colon and support homeostasis by metabolizing undigested polysaccharides into short-chain fatty acids and producing vitamins[15,16]. At the colonic mucosal barrier, which is constantly regenerated by fast-cycling LGR5-positive stem cells[17], dynamic microbiota-immune interactions take place[15,16]. Infection and chronic inflammation in the colonic mucosa can lead to inflammatory bowel disease (IBD)[18], which includes Crohn's disease and ulcerative colitis.

Embedded beneath the colonic mucosa are two ganglionated plexuses, the submucosal plexus and the myenteric plexus. This network of interconnected neurons composes the colonic enteric nervous system (ENS). Estimated to consist of over 100 million neurons in humans[19], the ENS regulates colonic motility, secretion, and vasodilation, performs various sensory functions, and interacts with the immune system. However, relatively little is known about the mechanisms by which colonic enteric neurons coordinate all these functions in real time[19–21]. Malfunction of the ENS can lead to a variety of conditions including a collection of maladies called functional gastrointestinal disorders (FGID), which includes irritable bowel syndrome, fecal incontinence, constipation, and others[18,20,21]. One in five people suffer from FGID, and pharmaceutical intervention is often unsuccessful partly due to the lack of fundamental understanding of the colonic ENS[22–25]. An emerging alternative intervention is peripheral nerve stimulation[26–30]. Sacral nerve stimulation (SNS) in particular has been shown to be an effective therapy for fecal incontinence and constipation[31–34]. However, little is known about the impact of SNS on the colonic ENS. Further, improving SNS efficacy is limited by the inability to directly visualize the SNS-ENS interaction in response to different stimulation parameters, especially at a single-cell resolution chronically in live animals[33–38].

We describe a colonic window that allows for chronic imaging of the same location in the colon. The custom 3D-printed titanium platform gives access to the colon while an implanted ferromagnetic scaffold provides stabilizing support underneath the colon to minimize movement during imaging. The magnetic implant hangs loose to minimize obstruction and inflammation when the animal is not being imaged. During imaging sessions, external magnets can be positioned to lift the scaffold and reversibly restrain the colon. Using this technology, we demonstrate real time imaging of colonic stem cells, immune cells, bacteria, and luminal content passage in live animals. Furthermore, using mice with transgenic calcium indicators, we observe spatiotemporal neuronal activity in the colon in response to SNS, showing direct evidence that SNS activates enteric neurons in vivo. This easy-to-use colonic implant technology will be a useful tool in furthering our understanding of colonic diseases.

## Results

**A colonic window with ferromagnetic scaffold.** To visualize the spatiotemporal dynamics of the colon, we designed a colonic window to span the abdominal cavity without restricting movement (Fig. 1a). One of the greatest challenges in imaging the colon is minimizing motion artifact caused by intrinsic motor patterns without obstructing gastrointestinal transit. To this end, we designed a custom ferromagnetic scaffold to implant beneath the colon with sufficient slack to allow free movement of the colon (Fig. 1b; Supplementary Fig. 1). The ferromagnetic scaffold was designed in AutoCAD 2016 (AutoDesk). The ferromagnetic scaffold can be manipulated under anesthesia non-invasively using external magnets to reversibly move and restrain the colon ventrally against the colonic window. Restraining the colon against the window magnetically reduces motion artifact over three-fold and allows for easier tracking of the same location in the colon (Fig. 1c, d; Supplementary Movie 1). The external magnets are removed after imaging sessions, which allows the colon to distend and function with minimal obstruction.

The colonic window is 3D-printed from smooth titanium to reduce scarring and improve biocompatibility, and heat-resistant UV fused-silica glass was laser-cut specifically to fit the custom window design. The window is designed to allow for unencumbered leg movement (Supplementary Movie 2) and unobstructed gastrointestinal transit, while providing a direct visual path to the colon. The target location in the colon was marked by tattoo for chronic imaging (Fig. 1e, f). The tattoo allows for easy visualization of shifts in colonic position from day to day, and it can be visualized directly during imaging. After recovering from surgery, mice move normally with no obvious indications of pain, discomfort, or impedance, as evaluated by institution veterinarians. This low-stress design allows for mice to walk around freely for weeks with the colon remaining unobstructed beneath the window. We confirmed there was no evidence of necrosis, degeneration, or lesions visible as a result of the procedure in hematoxylin and eosin slides prepared from animals which had window implants and tattoos for two weeks (Supplementary Fig. 2).

**Imaging of bacteria and luminal content in the colon in vivo.** Multiphoton microscopy is sufficient to image the entire colon wall from the serosa to the lumen. As proof-of-concept, we demonstrate live monitoring of fluorescent bacteria in the colon lumen in vivo. After fasting C57BL/6 mice for 3 h, we gavaged them with adherent invasive *Escherichia coli* (*E. coli*) derived from Crohn's disease patients and stably transformed to express a green fluorescent protein (GFP) tag[39]. Imaging via the colonic window shows that these bacteria were adherent to the colon crypts initially, although their presence declined between 12 and 36 h post gavage (Fig. 2a). For comparison, we simulated the passage of colon contents by gavaging Wnt1-tdTomato mice, after a three-hour fasting period, with a 70kD Oregon-Green dye and monitored the dye as it passed through the lumen of the colon. The fluorescent signal from the dye was clearly visible, displaying maximum fluorescence between 1.5 and 2 h post gavage (Fig. 2b, c). Colonic crypts were imaged using second harmonic generation microscopy. We confirmed the microscopy techniques have low autofluorescence by imaging through the colonic window in C57BL6/J mice (Supplementary Fig. 3). While second harmonic generation is possible in the non-transgenic mice, excitation with the utilized wavelength of 850 nm does not

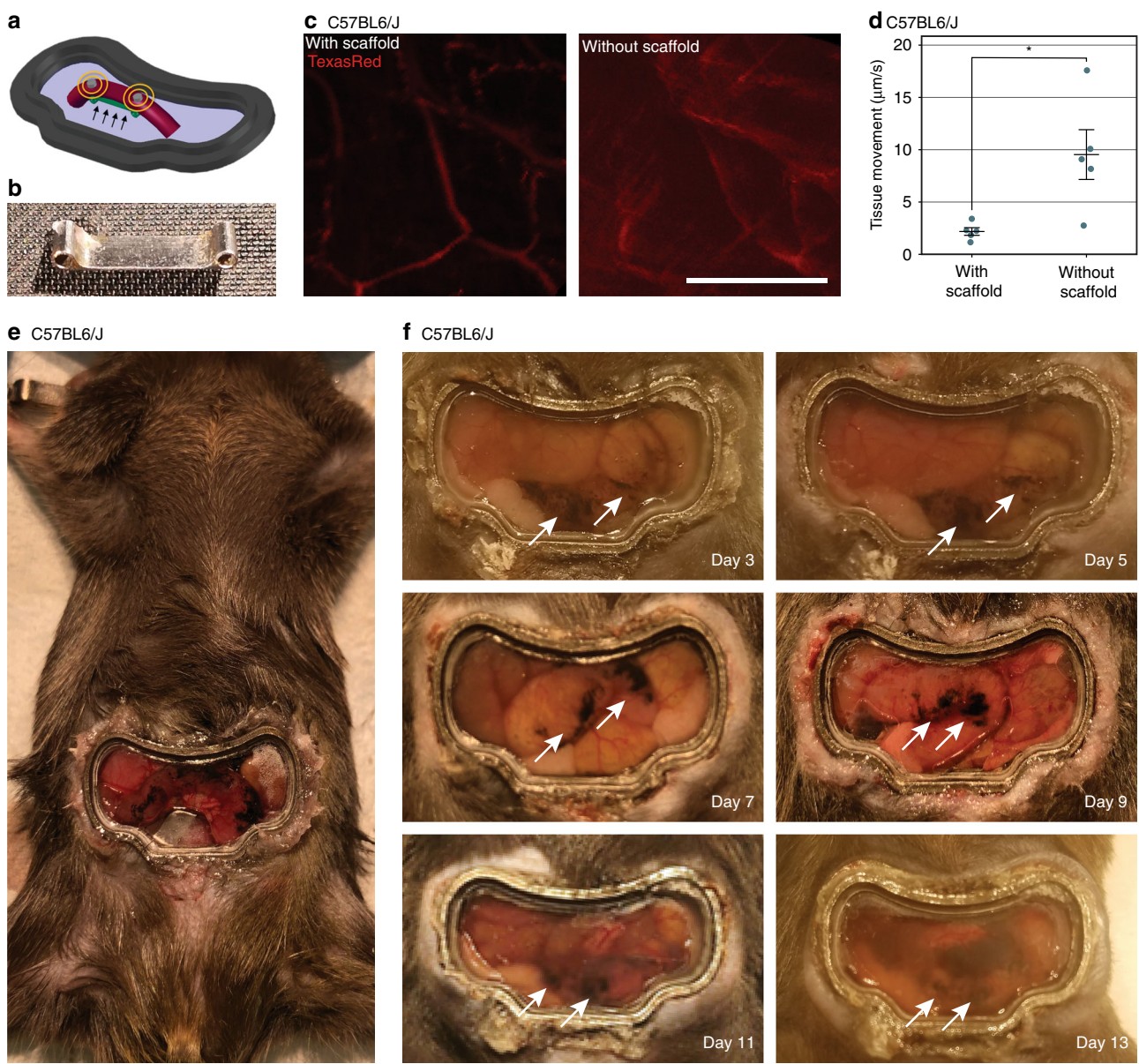

**Fig. 1 Colonic window with ferromagnetic scaffold for stabilized colon imaging. a** Diagram of the titanium colonic window with magnets placed on the surface above the ferromagnetic scaffold (green), located beneath the colon (red). The magnetic field (yellow) pulls the scaffold against the colon (arrows), holding the colon against the window during imaging. **b** Implantable ferromagnetic scaffold used to reversibly prevent colon movement during imaging. **c** Representative sum of 150 consecutive frames of vasculature (red) on the colon surface in a C57BL6/J mouse with and without the ferromagnetic scaffold activated by external magnets. **d** Quantification of tissue movement during time-series recordings of the colonic vasculature with and without the ferromagnetic scaffold activated by external magnets. Circles represent individual mice, and bars indicate $\mu \pm$ s.e.m. Star indicates a statistically significant difference by unpaired, two-tailed $t$-test ($n = 5$, $p < 0.05$). **e** Mouse implanted with the colonic window. **f** Colonic window implanted in a mouse for up to 13 days after implantation. Arrows indicate tattoo mark on the colonic wall to track position within the colon. All scale bars are 100 μm.

produce noticeable autofluorescence in the capture gate above 450 nm.

**Live monitoring of immune response to inflammatory challenge.** Next, we imaged immune cell activation in a live, chronic murine colitis model to visualize immune response in the colon. CX3CR1[GFP] x CCR2[RFP] mice were bred to label innate dendritic cells and inactivated monocytes (GFP) and activated monocytes (RFP)[40]. Mice underwent dextran sodium sulfate (DSS) challenge to induce inflammation[41,42]. Using multiphoton microscopy and the colonic window, we monitored CX3CR1-positive and CCR2-positive cells in the same location over time using the vasculature

as a roadmap. We monitored cells the day before DSS treatment, one day after treatment, and three days after treatment. We observed an increased number of activate, CCR2-positive monocytes in vivo within three days after treatment (Fig. 3a) as previously reported[43]. The vasculature landmarks labeled with Texas Red-Dextran consistently disappeared by the third day, likely due to leaky vasculature following inflammatory challenge. These methods demonstrate the colonic window enables tracking of monocyte recruitment and activation in response to colonic inflammation. We confirmed the increase of activated monocytes via conventional ex vivo imaging methods and flow cytometry analysis (Supp. Figure 4).

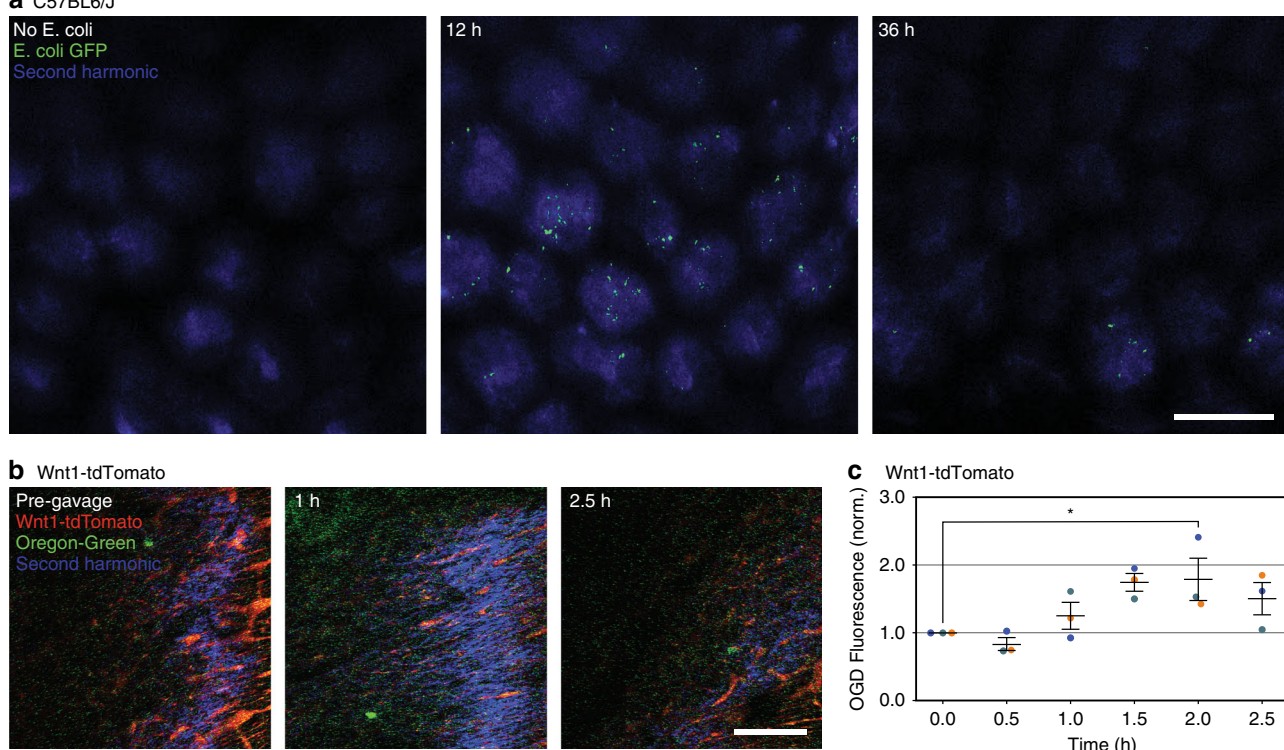

**Fig. 2 In vivo imaging of luminal content via the colonic window. a** Representative images from a C57BL6/J mouse gavaged with adherent invasive *E. coli* derived from a patient with Crohn's Disease and stably transformed to express GFP. Images taken from negative control, as well as 12 and 36 h after gavage of *E. coli* (green). Colonic crypts are visualized using second harmonic generation microscopy (blue). **b** Representative images from a Wnt1-tdTomato (red) mouse gavaged with Oregon-Green dye (green) and second harmonic generation microscopy (blue). **c** Quantification of Oregon-Green dye (OGD) fluorescence normalized to each subjects' pre-gavage (0 h) timepoint. Circles represent individual mice by color, and bars indicate μ ± s.e.m. Star indicates statistically significant difference from control by Dunnett's test for multiple comparisons after repeated-measures ANOVA ($n = 3$, $p < 0.05$). All scale bars are 100 μm.

**Tracking colon epithelial stem cell populations in vivo.** Regeneration of colonic epithelium is supported by the fast-cycling Lgr5-positive stem cells, which are reportedly radiation-sensitive[44,45]. Hyperplasia caused by these proliferating stem cells has also been associated with colorectal cancer[46–48]. To demonstrate that our intravital colonic imaging can capture the dynamics of these cells, we treated Lgr5-eGFP[17] mice with 18 Gy irradiation and tracked the colon over five days in the same mice. Intravital imaging reveals a clear decline in stem cell population following irradiation, consistent with the previously reported sensitivity to radiation[44,45] (Fig. 3b, c). We track the same location by administering 100 μL of 2.5% w/v Texas Red-Dextran by tail vein injection. Following irradiation in Lgr5-eGFP mice, we observed a statistically significant decrease in the size of the Lgr5-labeled crypts (Fig. 3d) and in the number of Lgr5-positive cells per crypt (Fig. 3e) in 5 mice. These observations demonstrate tracking of the same position in the colon for multiple days and a method to survey cell-type specific populations over time in response to injury.

**Sacral innervation and modulation in the colon.** The colonic ENS plays important roles in regulating motility and immune functions[19–21]. SNS is an emerging therapy for treating motility and immune disorders such as FGID and IBD[24,27–30,33–38]. However, the mechanisms by which SNS affects the ENS is largely unknown. Recently, Smith-Edwards and Najjar et al. demonstrated mouse colonic myenteric neurons respond to electrical stimulation of lumbosacral parasympathetic fibers under acute,

non-survival conditions[49]. Here, we used our intravital imaging technique to visualize the myenteric plexus in Wnt1- tdTomato mice over several hours through the colonic window and to test our ability to visualize this population while minimizing motion artifact (Supplementary Fig. 5). We also refined an image processing pipeline to track individual neurons to account for slight movements to further enhance quantification (Supp. Movie 3).

Next, we evaluated the effect of SNS on the firing properties of colonic myenteric neurons via genetically-encoded calcium indicators using Pirt-GCaMP3 mice[49]. Pirt-GCaMP3 mice were implanted with the colonic window and ferromagnetic scaffold. GCaMP3-fluorescence was visible in the colonic myenteric plexus through the colonic window (Fig. 4a). Pirt-GCaMP3 mice were also implanted with electrical stimulation leads in the S2 intervertebral foramen. Electrode implant location was confirmed by x-ray fluoroscopy (Supp. Fig. 6), and stimulating the sacral nerves evoked a motor response in the tail (Supp. Movie 4). Electrical stimulation amplitude was titrated to 20% below motor threshold. When electrical stimulation was delivered at 14 Hz, commonly used in FDA-approved therapeutic SNS[35,37,50], it evoked calcium responses in 10% of colon myenteric neurons on average in four out of five mice, with at least 25 cells per mouse (Fig. 4b; Supp. Movie 5). We repeated SNS at 7 Hz and 21 Hz, and we found that 14 Hz SNS evoked responses from a statistically significantly greater percentage of myenteric neurons than control conditions (Fig. 4c). When we collapsed across stimulation frequencies, we found the effect of SNS was not statistically significantly different from control (unstimulated) conditions (Fig. 4d).

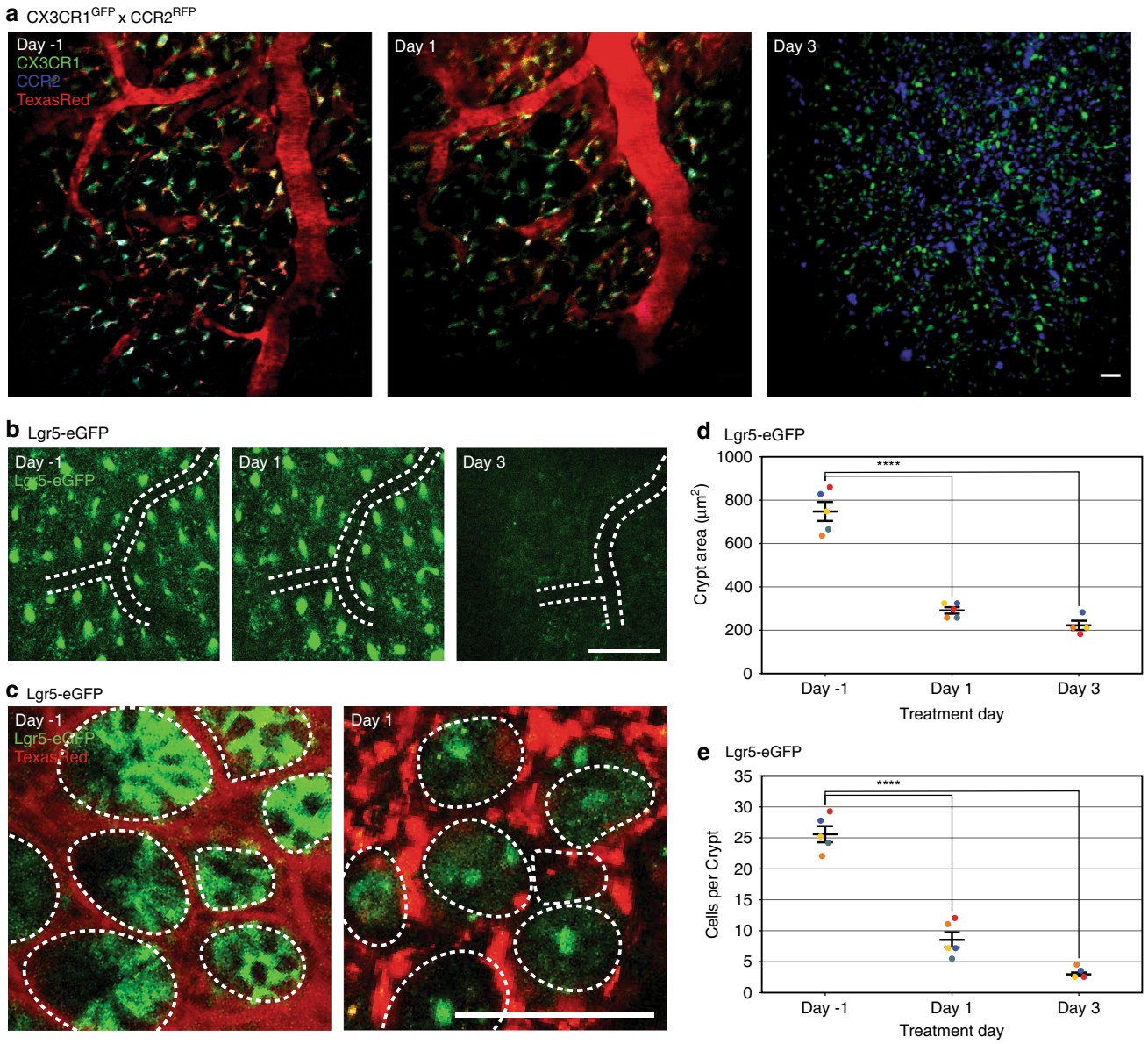

**Fig. 3 In vivo imaging of the same location over time via the colonic window. a** Representative images from a CX3CR1GFP × CCR2RFP mouse before and after DSS treatment (administered on Day 0). Using a combination of tattoo and vascular mapping we track innate dendritic cells and inactivated monocytes (green), activated monocytes (blue), and vasculature (red) in the same position for 5 days. **b** Representative images from a Lgr5-eGFP mouse before and after irradiation (treated on Day 0), demonstrating reduction in colonic crypts (green) after irradiation. Location is tracked by tattoo and vasculature branch points (white dashed lines) without vasculature dye. **c** Representative images from a Lgr5-eGFP mouse before and after irradiation with individual Lgr5-positive cells (green) and crypts labeled (white dashed lines). Location is tracked by tattoo and vasculature (red). **d**, **e** Quantification of Lgr5-positive **d**) crypt area ($\mu m^2$) and **e**) cells per crypt after irradiation (administered on Day 0). Circles represent individual mice by color, and bars indicate $\mu \pm$ s.e.m. Star indicates statistically significant difference from control by Dunnett's test for multiple comparisons after repeated-measures ANOVA ($n \geq 4$, $p < 0.001$). All scale bars are 100 $\mu$m.

We also traced the sacral nerves to compare their innervation of the lower gastrointestinal tract with the firing responses of colonic myenteric neurons. After injecting Fast Blue, a retrograde dye[51], into the S2 intervertebral foramen, we identified Fast Blue in the colonic myenteric plexus by imaging through the colonic window (Fig. 4e; Supplementary Fig. 7). Intravital imaging through the colonic window revealed Fast Blue-labeling in GCaMP-expressing myenteric neurons, which permitted us to monitor their calcium activity in real-time (Fig. 4f). On average, we observed Fast Blue labeling in 13.5% of GCaMP-expressing myenteric neurons in four mice (Fig. 4g). Of the cells labeled with

Fast Blue, 9.8% on average responded to SNS in four mice (Fig. 4h). These data demonstrate the colonic window can be used to monitor enteric neural activity in real time under chronic, survival conditions.

## Discussion

We demonstrate a colonic window is capable of imaging the colon chronically in live animals for up to two weeks. The image quality was dramatically improved by a ferromagnetic scaffold to stabilize the colon and minimize motion artifacts during imaging

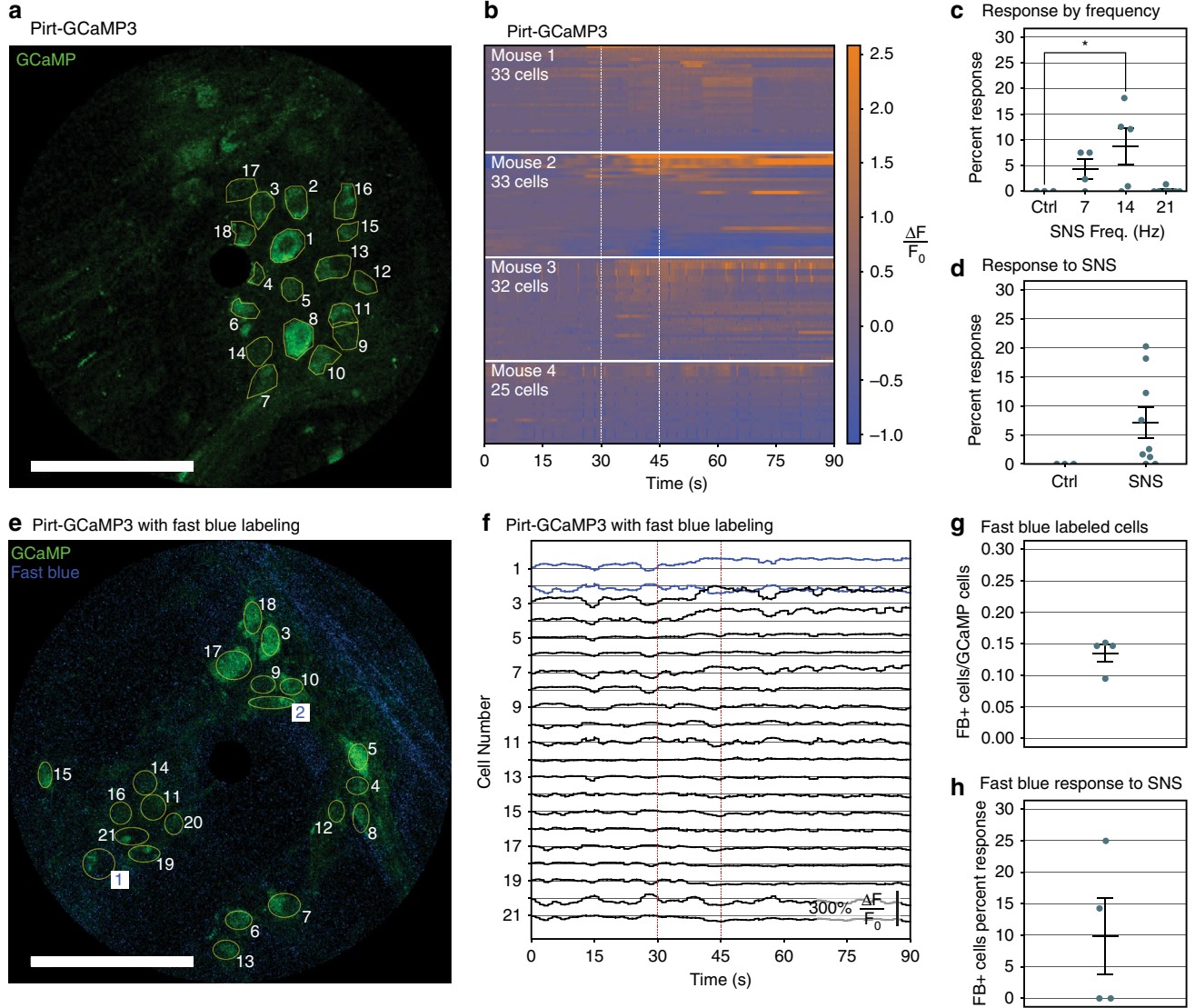

**Fig. 4 In vivo calcium imaging of the colonic myenteric plexus in response to SNS. a** Representative image from a Pirt-GCaMP3 mouse with GCaMP (green) expression in 18 outlined colonic myenteric neurons. **b** Calcium dynamics of at least 25 cells from four mice with 14 Hz SNS delivered from 30 to 45 s (vertical white dashed lines). Changes in GCaMP fluorescence are represented on a colorimetric scale. **c** Quantification of the percentage of GCaMP-positive cells that respond under control (ctrl, unstimulated) conditions, and with 7 Hz, 14 Hz, or 21 Hz SNS ($n \geq 3$). **d** The percent of GCaMP-positive cells that respond under control (ctrl, unstimulated) conditions and SNS collapsed across stimulation frequency. The increase in percent response with SNS is not significant by unpaired, two-tailed *t*-test ($n \geq 3$). **e** Representative image from a Pirt-GCaMP3 mouse injected with Fast Blue. GCaMP (green) expression is shown in 21 outlined colonic myenteric neurons, with Fast Blue (blue) found in 2 of the 21 cells (white/blue labels). **f** Changes in GCaMP fluorescence from **e** cells in response to 14 Hz SNS (vertical red dashed lines) in Fast-Blue-postive (blue) and -negative (black) cells. **g** Quantification of fraction of GCaMP-positive cells that are labeled with Fast Blue ($n = 4$). **h** Quantification of the percentage of Fast-Blue-positive cells that respond to SNS compared to all Fast-Blue-positive cells ($n = 4$). Circles represent individual mice, and bars indicate $\mu \pm$ s.e.m. Star indicates statistically significant difference from control by Dunnett's test for multiple comparisons after ANOVA ($p < 0.05$). All scale bars are 100 µm.

sessions without chronically obstructing gastrointestinal transit. We demonstrated a variety of applications of this technology by conducting in vivo imaging of bacteria, luminal content, immune cells, epithelial cells, and myenteric neurons. These results also validated key findings that, to our knowledge, have not been reported under chronic, survival conditions. We demonstrated live cell monitoring to track an immune response and stem cell loss during irradiation chronically in vivo in the same mice. Importantly, we also showed that SNS evokes calcium responses in colonic myenteric neurons repeatedly. However, it remains unclear how SNS modulates the ENS. Although it is possible the calcium responses are the result of direct, antidromic activation of enteric neurons projecting to the sacrum, this is unlikely

because the visceral sensory fibers are typically smaller in diameter and thus require much higher amplitude stimulation to evoke action potentials compared to larger diameter fibers. Nevertheless, our intravital imaging results support the importance of selecting stimulation frequency. The implantable colonic window has a large range of applications across a broad community of researchers to study various chronic, live interactions and processes in the colon with single-cell resolution for several days.

The field of intravital imaging has been expanding the range of organs that can be observed in vivo. However, each tissue poses unique challenges due to morphology and location. While alternatives such as endoscopic techniques can provide access to

certain regions, windows have three main benefits: a large field-of-view, compatibility with multiphoton imaging for deeper tissue penetration, and generating non-linear harmonic fluorescence from unlabeled tissue to provide structural context during imaging[52]. They also allow for easier accessibility of the target for manipulation through methods, such as optogenetic stimulation or laser ablation[11,53,54]. Previously, techniques have been developed to image moving tissues such as the beating of the heart[55] or the expansion of the lungs[8]. However, colonic motility patterns and large volume fluctuations has caused longstanding difficulties for imaging due to their highly variable nature[7,11]. Furthermore, small animal models are limited to extremely small and light-weight devices, making most battery-powered systems too bulky for implantation. Both of these issues were overcome with the ferromagnetic scaffold, which can reversibly restrain the colon without a battery. Physical stabilization via the ferromagnetic scaffold is supported by existing tools for image post-processing to track single cells, similar to methods used in the brain, spine and heart[55]. The combination of our colonic window design and ferromagnetic scaffold overcome the constraints of colon physiology and can be further expanded to other moving tissues and mouse models.

The capacity to image the colon in vivo repeatedly provides a window into the colonic dynamics of microbiota, stem cells, immune cells, myenteric neurons, and other cell types. Gut microbiota have been of great interest to the scientific community[56]. Microbiome research is expanding from studying bacteria cultures in an isolated environment to observing them as an integral part of the highly interactive host environment. For example, immune surface-pattern recognition receptors (such as toll-like and NOD-like receptors, etc.) can communicate with pathogen-associated molecular patterns expressed on the microbiota in order to keep the symbiotic bacteria thriving while containing pathogenic bacteria spread[57]. Murine models benefit from a large repository of transgenic lines. Dendritic cells[58,59], natural killer cells[60], monocytes[61], CD8 and CD4 T-cells[62,63], and others can be monitored using selective transgenic reported lines. The ability to observe how different subtypes of resident and circulating immune cells respond to challenges such as infection or inflammation in real time will benefit the scientific community[64].

The dynamics of intestinal LGR5-positive stem cells, which fuel epithelial regeneration, have been previously studied by intravital imaging[12,13]. These cells are susceptible to invasive bacteria and radioactivity[39,65]. APC mutation in these cells leads to expansion of the stem cell niche, giving rise to dysplasia and adenoma[48]. Our window allows imaging of such cells in the colonic epithelium, where colon cancer and some cases of IBD arise. The window can be incorporated into gastrointestinal disease models to study IBD[66,67], IBS[68], or colorectal cancer[69,70] to provide chronic, single-cell resolution in vivo. The imaging technique can be further coupled with in vivo CRISPR genome editing and endoscopic organoid transplantation to study colonic oncogenesis and metastasis[71].

Enteric neurons have been proposed to have a fast turnover rate compared to neurons in the central nervous system. This imaging technology can monitor enteric neuron development and turnover[72], which can be a useful technique for analyzing disease models like Hirschsprung's disease, to monitor aganglionosis in the distal colon[73,74]. While in vitro and ex vivo experiments provide useful data regarding ENS dynamics[75,76], it is more physiologically relevant to monitor the ENS in vivo in which extrinsic pathways are intact and the ENS can interact with immune cells, enteroendocrine cells, and others[49,77–80]. Such in vivo data will be essential for modeling the ENS in silico[79]. The window allows for unique insight into the effect of peripheral neuromodulation, as vagal and sacral nerve stimulation have therapeutic effects on gastrointestinal motility and inflammation[30,81–83].

## Methods

**Ethics statement**. All animal procedures were reviewed and approved by the Duke University Institutional Care and Use Committee (protocol A139–18–05 and 195–15–05) and Cornell University Institutional Care and Use Committee (protocol 2015–0029). They were conducted in accordance with the recommendations in the Guide for the Care and Use of Laboratory Animals[84] and all relevant regulations for animal testing and research.

All experiments were conducted on mice 6–12-weeks-old. Pirt-GCaMP3 mice were generated by targeted homologous recombination to replace the entire coding region of the Pirt gene with the GCaMP3 sequence fused with a neomycin resistance gene sequence and put in frame with the Pirt promoter,and Wnt1-cre:tdTomato (Wnt1-tdTomato) mice were generated by crossing Tg(Wnt1-cre)11Rth Tg(Wnt1-GAL4)11Rth/J (Jackson Laboratory: 003829) with Ai14 (Jackson Laboratory: 007914)[11]. Lgr5-EGFP-IRES-creERT2 mice were bred at Duke University (Jackson Laboratory 008875). Mice heterozygous for CX3CR1-EGFP (B6.129P(Cg)-Ptprca Cx3cr1tm1Litt/J; Jax # 005582) and CCR2-RFP (B6.129(Cg)-Ccr2tm2.1lfc/J; Jax # 017586) were bred in house by mating respective homozygous strains.

**Window design and implantation**. Designs for the window and the scaffold were created using AutoCAD 2016 (AutoDesk). The window frame was 3D-printed titanium (Materialise), and the window was cut from borosilicate heat-resistant UV fused -silica glass (Mark Optics, Corning 7980) with a 30 W $CO_2$ laser (Epilog, Zing). The ferromagnetic scaffold is $11.5 \times 3.5 \times 1.5$ mm. The scaffold was manufactured at the Duke Physics Instrumentation Shop using 416 ferromagnetic steel (AZO materials, UNS S41600).

All implants and surgical equipment were sterilized in an autoclave before use. Animals were anesthetized with isoflurane (4% vol/vol induction with 2–3% maintenance). Eye ointment was applied to prevent damage to the eyes. Mice were injected with bupivicane (0.25% < 2 mg/kg) and meloxicam (2 mg/kg once daily) subcutaneously prior to surgery. Surgical site was cleared using a shaving razor to remove hair and cleaned by alternating iodine and 70% ethanol treatments. A 15-mm diameter circle of the abdominal skin is surgically removed. The ferromagnetic scaffold was held in place by loops of suture passed through the anchor points and sutured to the abdominal wall on either side of the window, providing 2–3 mm of slack so the scaffold could be displaced during colonic distension or animal motion. The surgical site was marked on the surface of the colon intraoperatively with a tattoo using the AIMS tattoo system (Braintree Scientific, ATS-3 Q). The window is adhered to the surrounding tissue using Loctite 406 (Ellsworth, 135436) The success rate of reversible, consecutive, magnetic displacements was 100%, and mice survived past two weeks without any adverse events before reaching the experiment endpoint.

**Hematoxylin and Eosin histology**. The entire colon from a C57BL6/J mouse was harvested two weeks after window implant and tattoo. It was fixed in 10% neutral buffered formalin, embedded in paraffin, sectioned at 5 microns, stained with hematoxylin and eosin, and examined by an experienced veterinary pathologist (J. Everitt, Duke Research Animal Pathology Core). No evidence of necrosis or degeneration was noted and there were no significant lesions associated with the tattoo procedure.

**In vivo imaging**. Mice were anesthetized with isoflurane (4% vol/vol induction with 2–3% maintenance). Anesthetized mice were placed supine onto a 3D-printed imaging stage. Magnets (K&J Magnetics, D18) were positioned on top of the window to draw the ferromagnetic scaffold ventral within the abdominal cavity. Unless otherwise indicated, imaging was performed using a multiphoton microscope (Bruker Ultima IV, Zeiss LSM880, Olympus FV1000). At least one recovery day was allowed between imaging sessions. Freely-behaving animal movies were recorded with a GoPro HERO4 Silver.

**E. coli and fluorescent dye gavage**. C57BL/6 mice with colonic windows were gavaged with 100 µL adherent invasive E. coli-GFP (541–1, courtesy of Kenneth W. Simpson, BMV&S, PhD of Cornell University) grown overnight in 5 ml LB/ Ampicillin media from a single colony at 37 °C. Adherent invasive E. coli were imaged via the colonic window 12 and 36 h after gavage.

Wnt1-tdTomato mice were gavaged with 100 µL 2.5% w/v 70kD Oregon Green conjugated Dextran (ThermoFisher, D7176). In vivo imaging was performed at 30-minute intervals.

Second harmonic generation was performed with an excitation wavelength of 850 nm and a detection gate between 400–450 nm.

**Inflammation challenge with dextran sodium sulfate**. Mouse inflammation was induced by treating mice with 2% w/v DSS (Alfa Aesar, J63606) in drinking water for 3–6 days. We injected 100 µL 2.5% w/v Texas Red-Dextran (Thermofisher, D1863) retro-orbitally immediately prior to imaging to visualize the vasculature. Imaging was performed on a custom-built multiphoton microscope controlled by

the MATLAB program ScanImage 2018 (Pologruto et al., 2003). A Ti:Sapphire laser (Chameleon, Coherent) with the wavelength centered at 880 nm, was used to simultaneously excite GFP (CX3CR1), RFP (CCR2), and Texas-Red fluorescence (vasculature). Emission was separated using a 705 nm long-pass (LP) primary dichroic, a 593 nm LP secondary dichroic, a 538 nm LP tertiary dichroic and bandpass filters selective for RFP (575/25), Texas-Red (629/53) and GFP (<538). A water immersion ×20 objective (Olympus XLUMPlanFluor, 0.95 NA) was used. Z-stack images were collected to a depth of 100 μm from the adventitial layer (1 μm per step). Repeat imaging of the same region within the same mouse over subsequent days was performed by identifying the unique macroscopic vascular arrangement in addition to the microscopic features identified by fluorescence vascular labelling. In regions where DSS caused dramatic rearrangement of blood vessels at later time-points, the approximate matching region of the same mouse was imaged. RFP and Texas-Red have spectrally similar emission signals, therefore we used linear unmixing to identify GFP, RFP and Texas-Red signals. To determine the spectral contribution of each fluorophore, intensity ratios were calculated from regions of interest (ROI) manually selected based on location and morphology. Baseline images were used to select ROI judged to have high-GFP/low-RFP/low-Texas-Red with dendritic cell morphology or low-GFP/low-RFP/high-Texas-Red localized within the vascular lumen. Inflamed colon images from mice that did not receive intravenous Texas-Red were used to select ROI's judged to have low-GFP/high-RFP/low-Texas-Red and monocyte morphology.

**Ex vivo whole tissue imaging**. Segments of colon 1–2 cm long were collected from CX3CR1[GFP] × CCR2[RFP] mice. CX3CR1[GFP] x CCR2[RFP]intestine was opened longitudinally. The intestinal villi imaging was performed on a Zeiss Observer Z.1 microscope and an AxioCam MRm camera. Following intestinal villi imaging, flow cytometry analysis was done on single-cell suspension of digested colon tissue[85] using BD DiVa flow cytometer. The raw FACS data were analyzed with the FlowJo software to gate cells according to their forward (FSC) and side (SSC) scatter profiles and fluorescent intensities.

**Irradiation**. Five Lgr5-eGFP mice received 18 Gy total-body irradiation using an XRAD 320 irradiator (Precision X-ray Inc., North Branford, CT) four days after colonic window implantation. Irradiators are maintained and dosimetry was performed by the Duke University Radiation Safety Division staff. Unanesthetized mice were placed dorsally in a pie cage without restraints and with no shielding. All mice were placed 47.46 cm from the source and irradiated with 320 kVp, 12.5 mA X rays. The animals were irradiated using a dose rate of 2.23 Gy/min using a 2.0 mm Al filter. We located the same region in the colon during in vivo imaging as described using a tattoo or vascular dye. The tattoo location was used to guide alignment of the objective to the desired colon region. We visualized the vasculature to identify the same region in the colon by injecting 100 μL 2.5% w/v Texas Red-Dextran (Thermofisher, D1863) via tail vein immediately prior to imaging. We quantified the number of crypts labeled with GFP per time point averaged over 3 regions of colon captured in single planes.

**Sacral nerve electrode implantation and stimulation**. Two cardiac pacing electrodes (Medtronic, 6494) were implanted; the stimulation electrode was implanted in the S2 intervertebral foramen, and the return electrode was implanted in the subcutaneous tissue lateral and dorsal to the sacrum. The pacing electrodes were tunneled subcutaneously to a transcutaneous site at back of the neck.

Electrical stimulation was delivered with biphasic current-controlled pulses. The stimulation protocol was programmed with an arbitrary function generator (Tektronix, AFG1062) driving a biphasic stimulus isolator (Digitimer, DS4). Motor threshold was determined by gradually increasing current amplitude until a tail twitch was observed. Stimulation was performed at 20% below the recorded muscle-twitching threshold amplitude at 0.5% duty cycle.

**X-ray fluoroscopy**. Pirt-GCaMP3 mice were implanted with sacral nerve electrodes, as described. While under anesthesia, mice were imaged via X-ray fluoroscopy (GE, OEC 9800 Plus). Images were taken during mouse exhalation at 50% brightness, 60% contrast and recorded on compact disk and screen capture from the dorsal and lateral side.

**Nerve tracing**. Pirt-GCaMP3 mice were injected with 5 μL 4 mM FastBlue (Polysciences, 17740–1). The injections were made proximal to the S2 intervertebral foramen through an incision above the sacrum under isoflurane anesthesia. Fast Blue was visualized in vivo with multiphoton imaging.

**Image processing**. Tissue movement analysis was conducted in Imaris 8. Sphere tracking was used to identify cells (14-μm diameter, 15-μm step size, three-frame omission) in time series with and without the active magnetic implant (n = 5). The total distance traveled, and the time duration were averaged among the top 3 cells with the longest duration of continuous tracking in each frame.

Colon time series was performed in Wnt1-tdTomato mice, with z-stacks taken with 5-μm step size and depth from 210–425 μm, to accommodate for the shrinking or expanding colon between imaging sessions. Averaged z-projections and image rotation was performed in Fiji.

GCaMP images were processed on Imaris 8 (Oxford Instruments). We used sphere tracking (14-μm cell size, 20-μm max step movement, 5-frame omission) to account for cell movement pathways. We processed recordings using Fiji through StackReg-Rigid Body macro to remove micro frame shifts[86]. Cell regions-of-interest were drawn to identify active cells, and we used Time Series Analyzer V3 to record fluorescence intensity change. Repetitive breathing spike artifacts that cause single-frame (0.326 s) intensity spikes were averaged with surrounding frames to remove outliers.

**Statistical methods**. All statistical analyses were performed in JMP Pro 14 (SAS Institute). All tests used a significance value of 0.05. Here, we describe the statistical tests conducted and their specific use cases.

We used unpaired, two-tailed t-tests to evaluate the alternate hypothesis that two groups have unequal means. First, we compared tissue movement between imaging sessions with (n = 5) and without the magnetic scaffold (n = 5), and we found a statistically significant difference between the means of each group (Fig. 1d). We also used this method to compare the percentage of GCaMP-positive colonic myenteric neurons that responded to SNS (n = 9) to the percentage of responders in unstimulated conditions (n = 3). Responders are defined as cells that had a peak change in GCaMP fluorescence greater than 150% of the initial value, within a time period defined from the onset of stimulation to 15 s after stimulation. Independent samples required GCaMP fluorescence recordings in at least 17 unique GCaMP-positive cells per stimulation condition. One mouse was excluded from analysis because all GCaMP-positive cells failed to respond to any stimulus. The difference between the two means was not statistically significant (Fig. 4d).

We used repeated-measures ANOVA (RM-ANOVA) to evaluate the alternate hypothesis that repeated measurements in time have unequal means. In each of these cases, the same mice are used longitudinally throughout the study. First, we evaluated the effect of time on amount of visible Oregon-Green dye in Wnt1-tdTomato mice following gavage (n = 5). The Oregon-Green dye fluorescence at each time point was normalized to the initial value (pre-gavage) for each subject. RM-ANOVA revealed a statistically significant effect of time, with subject included as a random variable. We made multiple comparisons between each post-gavage time point to the pre-gavage time point using Dunnett's test for multiple comparisons. Dunnett's test revealed a statistically significant difference between the mean fluorescence pre-gavage and the mean fluorescence 2 h after gavage (Fig. 2c). We used the same method, RM-ANOVA and Dunnett's test for multiple comparisons, to evaluate the effect of time on the crypt area and number of cells per crypt in Lgr5-eGFP mice after irradiation. Five mice were used in the study, although one in five met the humane endpoint prior to Day 3 after irradiation. The RM-ANOVA and Dunnett's test revealed as statistically significant difference between the crypt area (Fig. 3d) and number of cells per crypt (Fig. 3e) one day and three days after irradiation, compared to one day before irradiation in the same mice.

We used a one-way ANOVA to evaluate the alternate hypothesis that different stimulation conditions have unequal means when measuring the percent responders among GCaMP-positive cells. Independent samples required GCaMP fluorescence recordings in at least 13 unique GCaMP-positive cells per stimulation condition. We compared the mean percent response in unstimulated (control) conditions (n = 3), and SNS at 7 Hz (n = 4), 14 Hz (n = 5), and 21 Hz (n = 7). One mouse was excluded from analysis because all GCaMP-positive cells failed to respond to any stimulus. One-way ANOVA revealed a statistically significant effect of stimulation condition. We made multiple comparisons between each SNS condition to the unstimulated control using Dunnett's test for multiple comparisons. Dunnett's test revealed a statistically significant difference between SNS at 14 Hz and unstimulated controls (Fig. 4c).

**Reporting summary**. Further information on research design is available in the Nature Research Reporting Summary linked to this article.

# Data availability

The authors declare that the data supporting the findings of this study are available within the paper and its supplementary information files or available from the corresponding author upon reasonable request, including raw (uncompressed) image files for the data in Figs. 1c, d, e, f, 2, 3, 4.

# Code availability

The authors declare that the custom code used for data processing and analysis in this study are available from the corresponding author upon reasonable request. The data in Fig. 4 have associated custom code.

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

## Acknowledgements

We thank Andrea Nackley, PhD of Duke University for providing us with hardware for ex vivo tissue imaging experiments, Roberto Manson, MD of Duke University for assistance with x-ray fluoroscopy, Jeffrey Everitt, DVM of Duke University for histo-pathological preparation and analysis, and Kenneth Simpson, BMV&S, PhD of Cornell University for providing us with adherent invasive E. coli bacteria. We also express gratitude to the staff of the Light Microscopy Core Facility, Flow Cytometry Core Facility, and the Innovation Co-Lab at Duke University. This work was supported by the National Institutes of Health (R35GM122465 and R01DK119795 to X.S.), the Defense Advanced Research Projects Agency (N66001-15-2-4059 to X.S.), and the National Cancer Institutes (R35CA197616 to D.G.K.). Imaging data was acquired through the Cornell University Biotechnology Resource Center, with NYSTEM (CO29155) and NIH (S10OD018516) funding for the shared Zeiss LSM880 confocal/multiphoton microscope and through the Duke University Light Microscopy Core Facility.

## Author contributions

N.R., A.G., S.D., B.B.B., and X.S. designed the experiments and wrote the manuscript. N.R. and A.G. performed the experiments. K.C. helped design and conduct surgical procedures and tattooing. C.E., D.M.S., M.A.M., and N.N. helped design and conduct the DSS challenge and associated in vivo imaging experiments. A.R.D. and D.G.K. helped design and conduct the irradiation challenge. M.M.K. and D.V.B. assisted with the design and conduction of intravital calcium imaging experiments. Q.H. assisted with the intravital imaging of Lgr5-positive cells. A.G. and B.B.B. analyzed the data.

## Competing Interests

The authors declare no competing interests.
