## [Peer Review File · Nature Communications]

Reviewers' comments:

Reviewer #1 (Remarks to the Author):

The manuscript by Rakhilin et al. describes their successful adaptation of abdominal window technology to image colonic motility in mice. This is an interesting approach that builds further on published work, correctly cited, by the group of Van Rheen et al. They present an abdominal window which enables chronic imaging of the colon in live mice. The authors introduce a ferromagnetic scaffold implant which facilitates chronic imaging by reducing artefacts due to colonic contractile activity. In the absence of external magnets, the scaffold allows the gut to freely contract when the animal is not being imaged thereby minimizing obstruction. The feasibility and broad applicability of this technique was demonstrated through intravital imaging of GFP-labelled *E. coli* in the colonic crypts, passage of an orally-gavaged fluorescent dye, colonic GFP-expressing LGR5+ stem cells, fluorescent immune cells, and Ca²⁺ activity of enteric neurons following electrical sacral nerve stimulation (SNS). This paper is a follow-up of their own small intestinal imaging work published earlier. The authors chose to list a number of examples of what they are and will be able to do with this technology, which unfortunately reads a bit as a collection of individual proof-of-principle examples, not all of them recorded *in vivo*. Below are a number of issues, questions and comments that should be addressed to improve the paper.

1/ Throughout the paper it should be made clear what the statistical n-numbers actually represent. For instance in figure 2c, is the SEM (green shadow) calculated from different animals, different repeats of the gavage? And how come there is no variation half an hour after gavage? Similarly, in figure 4 a and f. Is the variation depicted in the shaded plots, based on different cells, different recordings or different mice?

2/ Some of the data presented in this paper are not relevant to the technical focus of the paper. For instance why exactly was capsaicin used in *ex-vivo* preparations of the gut otherwise some irrelevant *ex-vivo* controls that do not underscore the abdominal window approach in this paper. It is not clear how the *ex-vivo* capsaicin experiments "validate the neuronal calcium activity".

3/ The passage of luminal contents were examined by gavaging mice with a fluorescent oregon-green dye. Were the mice first fasted prior to introducing the fluorescent dye? If not, how does potential autofluorescence from the luminal content already present in the gut affect imaging of the fluorescent dye?

4/ In the results, the authors state that they "observed an increased number of activated RFP-labeled monocytes flocking to the gut *in vivo* (Fig. 2d, Supp. Fig 1a)..." Figure 2d appears to show presumably only a comparison of the colon of a control vs. DSS-treated mouse. This Figure panel also gives the false impression that this was recorded in a live mouse. The methods concerning the dual reporter (CX3CR1-GFP CCR2-RFP) mice however only mention *ex vivo* recordings. This is misleading. A comparison (and ideally a quantification) of the CXCR1-GFP and CCR2-RFP signal before, and over the course of DSS-treatment in the same mouse/gut region are necessary to support this statement. In addition, it would be helpful to provide a description of the time point/s at which the images of the control and DSS-treated colon/s were taken to show the progression of the "flocking" of RFP+ monocytes to the gut. Moreover, supplementary figure 1a refers to *ex vivo* rather than *in vivo* data – this is not obvious in the text presented.

5/ The authors should provide all information about the ferromagnetic insert. What were the dimensions and design. What happens after imaging when the magnets are removed is the insert going back to its original position? How was it kept in place? What was the success rate of consecutive magnetic displacement (100%)?. How long did mice survive with a ferromagnetic part connected to the colon? It's mentioned that the part was sutured to the skin? What does that mean?

6/ The suppl. movies 8-10 are intriguing as they suggest that a patch of neurons is responding to the sacral nerve stimulation. Was this consistent over different mice?

7/ In figure 2, the authors mention "second harmonic crypt fluorescence", what do they mean by

that? Second harmonic generation does not have anything to do with fluorescence. Please amend. Moreover the images in suppl. figure 2 show realistic second harmonic collagen structures, however the middle panel of Figure 2a, it appears as mainly autofluorescence was recorded. What was the wavelength of incoming light and how were the second harmonic signals detected?

8/ The authors describe a means of marking a particular location with tattoos for chronic imaging, but overall its applicability and practical utility is unclear. How effective was this approach for relocation of the same region and for repeated imaging of the same region? How does the tattoo labelling itself affect the marked region of gut? The images in figure 1f are too small to be informative. Where does the change in color come from? On day 11 it is not clear what is the ink of the tattoo or whether there is some necrosis already.

9/ In the discussion the authors state that their post-processing image stabilization provides a platform for other cell tracking. Without providing the necessary software tools, this is a claim that cannot be made. Several different groups have, with varying success, developed tracking and image registration tools some of them commercially available as in Imaris or in open access platforms like ImageJ.

10/ In the videos provided about mouse behavior, the mouse does not appear to be very explorative and agile. Was this the normal behavior after the surgery? Did the animal facility vet or caretaker approve that this behavior was within the normal (unpainful) range?

11/ Can the authors explain what happens during the electrical stimulation of the sacral nerve? The retrograde tracing experiment show that some neurons in the colon project to the DRG's. When stimulating the sacral nerves, action potentials would back propagate into the neurons in the intestinal wall. Thus, are these enteric neurons directly stimulated? Or does that stimulation work in another way?

12/ In supplementary figure 5 three out of four channels completely overlap: a nuclear staining, a microtubular label (Tuj1) and a retrograde tracer (mostly appearing a bit vesicular). This is impossible and clearly an artifact. This should not have escaped the attention of the authors.

13/ E. coli that were isolated from Crohn's patients 'were labeled with GFP'. What do the authors mean by that? Were the bacteria stably transformed such that they expressed GFP?

14/ It is unclear how the imaging of GFP-LGR5+ stem cells were performed. Were images taken from a single plane or are these maximum projections of the crypts taken at multiple planes? Is it possible that some cells were missed simply because they were out of the focal plane imaged if the surface of the gut imaged was not completely flat? Further, is it possible to track or repeatedly image the same intestinal region over time (using the tattoos)? The axis title of supplementary figure 2 is redundant: "...visible crypts visible".

15/ The authors describe the imaging of the colonic ENS using Wnt1-cre:tdTomato mice in the results but the purpose of this data in the context of testing the effect of SNS on colonic neuronal firing is unclear. Would it not be more informative to show the GCaMP3 signal in the ENS over time?

16/ The data presented in Fig 4f illustrating Ca²⁺ activity in the colonic myenteric neurons following SNS is only shown for the different frequencies tested, but lacks a comparison with baseline activity without stimulation over a 2 min recording.

17/ Given the effectiveness of fast blue labelling of the sacral innervation of the colon, is it possible to examine the Ca²⁺ responses of FB-labelled enteric neurons to SNS? This may provide stronger support for the SNS-evoked activation enteric neurons.

18/ Animals were imaged for several days in a row. Some up to 11 days? With neuron turnover rate as suggested by Kulkarni et al. (cited as reference 74), the authors should have seen around a 60% renewal of neurons. Was this indeed the case?

19/ A DSS model should not be called an IBD model. DSS causes colitis indeed, but it does not

necessarily mimic

Reviewer #2 (Remarks to the Author):

Dear Editor !
Dear Dr. Shen !
Dear Authors !

Thank you for giving me the opportunity and honor to review an article for the important journal "Nature Communications".

It was absolutely a pleasure to read this highly innovative study. The technique is spectacular !!!
The manuscript is well written and you can follow every step easily.
The author do not only present the new experimental setup up, they also show various different applications. The main figures, supplementary figures and videos are brilliant !
The efforts to conduct this wonderful study have to be "ginormous" !

The only point the author may clarify "the custom-designed implant was inserted under the colon and sutured to the skin" - do the thread run from the borders of the colon to the skin at the border of the window ?!

I highly recommend to publish the manuscript in the present form without any modifications !

I am absolutely excited regarding the "new window" which is now open for researcher to study the colon in vivo !

Yours sincerely !

Philipp Lenz, M.D.

Reviewer #1

1. Throughout the paper it should be made clear what the statistical n-numbers actually represent. For instance, in figure 2c, is the SEM (green shadow) calculated from different animals, different repeats of the gavage? And how come there is no variation half an hour after gavage? Similarly, in figure 4 a and f. Is the variation depicted in the shaded plots, based on different cells, different recordings or different mice?

We thank the reviewer for pointing out this ambiguity and have made extensive changes to the text and figures in order to more clearly convey biological versus technical replicates throughout. All quantification in the revisions depict individual values for each mouse (independent sample), as well as the mean and standard error of the mean. This convention is used in the revised **Fig. 1d**, **Fig. 2c**, **Fig. 3d,e**, and **Fig. 4c,d,g,h**. The statistical methods section has been updated (**p. 16**) to explicitly clarify what constitutes an independent sample and how many independent samples are included in each statistical test. We also clarify the statistical tests used.

In **Fig. 2c**, we have added distinct data points for each animal (by color) in the quantification plot and used a more conventional visualization of the mean and SEM of the complete dataset.

Fig. 4 has been reworked extensively in order to more clearly and completely report our data. Rather than providing a bulk response profile in a representative trace, we have generated a heat map (**Fig. 4b**) from multiple animals as is commonly used to report calcium imaging results. We also show all calcium traces from a single representative field-of-view (**Fig. 4f**). We hope the reader can more easily see the variation in response (or lack thereof) within the population of observed cells across animals. Bulk statistical analysis has been separated out for ease of comprehension.

2. Some of the data presented in this paper are not relevant to the technical focus of the paper. For instance, why exactly was capsaicin used in ex-vivo preparations of the gut otherwise some irrelevant ex-vivo controls that do not underscore the abdominal window approach in this paper. It is not clear how the ex-vivo capsaicin experiments “validate the neuronal calcium activity”.

We appreciate the feedback and agree with the reviewers that *ex vivo* experiments do not directly support the novel colon window technology as the focus of this paper. The *ex vivo* calcium imaging has been removed.

3. The passage of luminal contents were examined by gavaging mice with a fluorescent Oregon-Green dye. Were the mice first fasted prior to introducing the fluorescent dye? If not, how does potential autofluorescence from the luminal content already present in the gut affect imaging of the fluorescent dye?

We fasted mice for 3 hours before gavage. Additionally, we demonstrate low baseline autofluorescence by taking a pre-gavage image of the same animal and imaging in C57BL6/J mice without fluorescent reporters (**Fig. 2b**; **Supp. Fig. 3**).

We amended our manuscript to more clearly reflect the methodology used in this study:

“After fasting C57BL/6 mice for three hours, we gavaged them with adherent invasive *Escherichia coli*...” (p. 5, § Imaging of bacteria and luminal content in the colon *in vivo*).

“For comparison, we simulated the passage of colon contents by gavaging Wnt1-tdTomato mice, after a three-hour fasting period, with a 70kD Oregon-Green dye and monitored the dye as it passed through the lumen of the colon” (p. 5, § Imaging of bacteria and luminal content in the colon *in vivo*).

“We confirmed the multiphoton microscopy techniques have low autofluorescence by imaging through the colonic window in C57BL6/J mice (Supp. Fig. 3)” (p. 5, § Imaging of bacteria and luminal content in the colon *in vivo*).

4. In the results, the authors state that they “observed an increased number of activated RFP-labeled monocytes flocking to the gut *in vivo* (Fig. 2d, Supp. Fig 1a)...” Figure 2d appears to show presumably only a comparison of the colon of a control vs. DSS-treated mouse. This Figure panel also gives the false impression that this was recorded in a live mouse. The methods concerning the dual reporter (CX3CR1-GFP CCR2-RFP) mice however only mention *ex vivo* recordings. This is misleading. A comparison (and ideally a quantification) of the CXCR1-GFP and CCR2-RFP signal before, and over the course of DSS-treatment in the same mouse/gut region are necessary to support this statement. In addition, it would be helpful to provide a description of the time point/s at which the images of the control and DSS-treated colon/s were taken to show the progression of the “flocking” of RFP+ monocytes to the gut. Moreover, supplementary figure 1a refers to *ex vivo* rather than *in vivo* data – this is not obvious in the text presented.

We have made changes to both the text and figure legends to clarify how data were obtained.

“Mice underwent dextran sodium sulfate (DSS) challenge to induce inflammation⁴¹.⁴² Using multiphoton microscopy and the colonic window, we monitored CX3CR1-positive and CCR2-positive cells in the same location over time using the vasculature as a roadmap. We monitored cells the day before DSS treatment, one day after treatment, and three days after treatment. We observed an increased number of activated, CCR2-positive monocytes *in vivo* within three days after treatment (Fig. 3a) as previously reported⁴³” (p. 6, § Live monitoring of immune response to an inflammatory challenge in the colon *in vivo*).

The content originally comprising **Fig. 2d** is now **Fig. 3a**, and it has been revised to show images taken through the window from the same location in the same mouse over a five-day period. We label each inset with the day the image was taken relative to treatment onset (i.e. treat on day 0).

The representative images from this study are useful in highlighting the potential utility of the window in tracking multiple types of cells resident to the colon, but our sample size for this study was limited because we no longer maintain this mouse strain and the strain was not maintained when the lab moved from Cornell to Duke. We had to borrow the few remaining ones from another lab back at Cornell. Therefore, we include representative images before and after DSS treatment, but refrain from making claims about the magnitude of effect due to low statistical power.

However, we do include a more thorough quantification of Lgr5-eGFP mice in response to irradiation because we do maintain this mouse strain (**Fig. 3d,e**).

5. The authors should provide all information about the ferromagnetic insert. What were the dimensions and design? What happens after imaging when the magnets are removed is the insert going back to its original position? How was it kept in place? What was the success rate of consecutive magnetic displacement (100%)? How long did mice survive with a ferromagnetic part connected to the colon? It's mentioned that the part was sutured to the skin? What does that mean?

We have revised the methods and results to more thoroughly describe the scaffold composition, dimensions, manufacture, and use, as it is integral to the functionality of this technique. We have also revised **Fig. 1b** so that readers can more clearly see the physical insert, and we added a 3D-render of the CAD design from multiple angles (**Supp. Fig. 1**).

From the Methods and Materials, **p. 11,12**, § Window Design and Implantation:

Designs for the window and the scaffold were created using AutoCAD 2016 (AutoDesk). The window frame was 3D-printed titanium (Materialise), and the window was cut from borosilicate heat-resistant UV fused -silica glass (Mark Optics, Corning 7980) with a 30 W CO₂ laser (Epilog, Zing). The ferromagnetic scaffold is 11.5 x 3.5 x 1.5 mm. The scaffold was manufactured at the Duke Physics Instrumentation Shop using 416 ferromagnetic steel (AZO materials, UNS S41600).

All implants and surgical equipment were sterilized in an autoclave before use. Surgery was conducted on adult mice using sterile technique, isoflurane anesthesia, and analgesics as described previously¹¹. The ferromagnetic scaffold was held in place by loops of suture passed through the anchor points and sutured to the abdominal wall on either side of the window, providing 2-3 mm of slack so the scaffold could be displaced during colonic distension or animal motion. The surgical site was marked on the surface of the colon intraoperatively with a tattoo using the AIMS tattoo system (Braintree Scientific, ATS-3 Q). The success rate of reversible, consecutive, magnetic displacements was 100%, and mice survived past two weeks without any adverse events before reaching the experiment endpoint.

6. The suppl. movies 8-10 are intriguing as they suggest that a patch of neurons is responding to the sacral nerve stimulation. Was this consistent over different mice?

Yes, we were able to replicate a response to SNS stimulation in a subset of neurons in multiple mice. We believe this should be more readily apparent with our changes to **Fig. 4b**, which includes a heat map with clearly delineated recordings from distinct cells in four Pirt-GCaMP3 mice. The quantification in **Fig. 4c,d,g,h** show unique animals marked by an independent datapoint.

7. In figure 2, the authors mention “second harmonic crypt fluorescence”, what do they mean by that? Second harmonic generation does not have anything to do with fluorescence. Please amend. Moreover the images in suppl. figure 2 show realistic second harmonic collagen structures, however the middle panel of Figure 2a, it appears as mainly autofluorescence was recorded. What was the wavelength of incoming light and how were the second harmonic signals detected?

We have revised the text, figures, and figure captions to clarify our use of second harmonic generation microscopy. We included new images from C57BL6/J mice without endogenous fluorescence to clearly distinguish between channels and demonstrate low levels of autofluorescence during second harmonic generation microscopy (**Supp. Fig. 3**).

“Figure 2. In vivo imaging of luminal content via the colonic window. (a) Representative images from a C57BL6/J mouse gavaged with adherent invasive *E. coli* derived from a patient with Crohn’s Disease and stably transformed to express GFP. Images taken from negative control, as well as 12 and 36 hours after gavage of *E. coli* (green). Colonic crypts are visualized using second harmonic generation microscopy (blue). **(b)** Representative images from a Wnt1-tdTomato (red) mouse gavaged with Oregon-Green dye (green) and second harmonic generation microscopy (blue). **(c)** Quantification of Oregon-Green dye (OGD) fluorescence normalized to each subjects’ pre-gavage (0 hour) timepoint. Circles represent individual mice by color, and bars indicate $\mu \pm$ s.e.m. Star indicates statistically significant difference from control by Dunnett’s test for multiple comparisons after repeated-measures ANOVA ($n = 3$). All scale bars are 100 μm .” (**Fig. 2** caption).

“Colonic crypts were imaged using second harmonic generation microscopy. We confirmed the microscopy techniques have low autofluorescence by imaging through the colonic window in C57BL6/J mice (Supp. Fig. 3)” (**p. 5**, § Imaging of bacteria and luminal content in the colon *in vivo*).

“Second harmonic generation was performed with an excitation wavelength of 850 nm and a detection gate between 400-450 nm” (**p. 13**, § *E. coli* and Fluorescent Dye Gavage).

8. The authors describe a means of marking a particular location with tattoos for chronic imaging, but overall its applicability and practical utility is unclear. How effective was this approach for relocation of the same region and for repeated imaging of the same region? How does the tattoo labelling itself affect the marked region of gut? The images in figure 1f are too small to be informative. Where does the change in color come from? On day 11 it is not clear what is the ink of the tattoo or whether there is some necrosis already.

We thank the reviewer for their questions regarding this technique, and we have made several changes to better demonstrate use of this tool. In **Fig. 1f**, we rearranged the panels to allow more room for the visualization of the tattoo over multiple days. We attribute color change to different lighting conditions, which we made more consistent in the new series of images. Some color change can also be attributed to normal fading of the tattoo following recovery from the window implant procedure. To address concerns of necrosis or other technique related abnormality, we have also added a new H&E histological study which demonstrates lack of pathology in these animals as evaluated by a certified veterinary pathologist in **Supp Fig. 2**.

“We confirmed there was no evidence of necrosis, degeneration, or lesions visible as a result of the procedure in hematoxylin and eosin slides prepared from animals which had window implants and tattoos for two weeks (**Supp. Fig. 2**)” (p. 5, § A colonic window with ferromagnetic scaffold).

“The entire colon from a WT B6/J mouse was harvested two weeks after window implant and tattoo. It was fixed in 10% neutral buffered formalin, embedded in paraffin, sectioned at 5 microns, stained with H&E and examined by an experienced veterinary pathologist (J. Everitt, Duke Research Animal Pathology Core). No evidence of necrosis or degeneration was noted and there were no significant lesions associated with the tattoo procedure.” (p. 12, § Hematoxylin and Eosin Histology).

9. In the discussion the authors state that their post-processing image stabilization provides a platform for other cell tracking. Without providing the necessary software tools, this is a claim that cannot be made. Several different groups have, with varying success, developed tracking and image registration tools some of them commercially available as in Imaris or in open access platforms like ImageJ.

We have amended the discussion for clarity to ensure we do not make unsupportable claims. We did not write any custom/specialized software; rather, we simply used a combination of commercially available tools through the Imaris and Fiji platforms to preform our stabilization and analysis, and provide details related to this process in the methods should others wish to replicate the results. While we are not aware of any group which has used our method to stabilize images before, our focus is on the abdominal window rather than the post-processing techniques, and we only wish to

be clear regarding any post-processing performed and provide details on how readers can replicate our methods.

“Physical stabilization via the ferromagnetic scaffold is supported by existing tools for image post-processing to track single cells, similar to methods used in the brain, spine and heart⁵⁵. The combination of our colonic window design and ferromagnetic scaffold overcome the constraints of colon physiology and can be further expanded to other moving tissues and mouse models.” (p. 9, § Discussion).

10. In the videos provided about mouse behavior, the mouse does not appear to be very explorative and agile. Was this the normal behavior after the surgery? Did the animal facility vet or caretaker approve that this behavior was within the normal (unpainful) range?

Facility veterinarians have approved that our animals do not experience impaired mobility or demonstrate signs of pain following a standard 48-hour recovery period wherein we administer an analgesic as we would after any surgery. We have amended the manuscript to reflect this.

“After recovering from surgery, mice move normally with no obvious indications of pain, discomfort, or impedance, as evaluated by institution veterinarians” (p.5, § A colonic window with ferromagnetic scaffold).

11. Can the authors explain what happens during the electrical stimulation of the sacral nerve? The retrograde tracing experiment show that some neurons in the colon project to the DRG's. When stimulating the sacral nerves, action potentials would back propagate into the neurons in the intestinal wall. Thus, are these enteric neurons directly stimulated? Or does that stimulation work in another way?

A recent study by Smith-Edwards and Najjar et al. [recently e-published after our initial submission, now cited in the text as #49] suggest electrical stimulation of the sacral nerves is activating colonic myenteric neurons synaptically. It is possible antidromic activation could evoke calcium responses in enteric neurons, but this is unlikely because the visceral sensory fibers require greater current amplitude to evoke a response compared to other fibers.

While the tracing experiment indicates that some colonic neurons project to the DRGs, these are likely small fiber diameter neurons and thus more difficult to evoke a response in using extracellular stimulation like that in our SNS preparation.

We expect SNS is activating larger diameter fibers in the sacral nerves whose processes project to the colon (and thus would not be labeled by a retrograde tracer injected into the sacrum). It is likely that these fibers form synapses with enteric neurons and activate the descending pathway similar to what has been reported by Smith-Edwards and Najjar.

The following text was added to the Discussion on p. 8,9:

Importantly, we also showed that SNS evokes calcium responses in colonic myenteric neurons repeatedly. However, it remains unclear how SNS modulates the ENS. Although it is possible the calcium responses are the result of direct, antidromic activation of enteric neurons projecting to the sacrum, this is unlikely because the visceral sensory fibers are typically smaller in diameter and thus require much higher amplitude stimulation to evoke action potentials compared to larger diameter fibers. Nevertheless, our intravital imaging results support the importance of selecting stimulation frequency.

12. In supplementary figure 5 three out of four channels completely overlap: a nuclear staining, a microtubular label (Tuj1) and a retrograde tracer (mostly appearing a bit vesicular). This is impossible and clearly an artifact. This should not have escaped the attention of the authors.

We appreciate the reviewer's attention to detail. In hindsight we agree that this figure is not of sufficient quality or clarity for inclusion in this or any other manuscript. As the work was completed in post-mortem sections and does not support the functionality of the window we have decided to remove it from this work.

13. E. coli that were isolated from Crohn's patients 'were labeled with GFP'. What do the authors mean by that? Were the bacteria stably transformed such that they expressed GFP?

We have amended the manuscript text.

"After fasting C57BL/6 mice for three hours, we gavaged them with adherent invasive Escherichia coli (E. coli) derived from Crohn's disease patients and stably transformed to express a green fluorescent protein (GFP) tag³⁹" (p.5, § Imaging of bacteria and luminal content in the colon *in vivo*).

14. It is unclear how the imaging of GFP-LGR5+ stem cells were performed. Were images taken from a single plane or are these maximum projections of the crypts taken at multiple planes? Is it possible that some cells were missed simply because they were out of the focal plane imaged if the surface of the gut imaged was not completely flat? Further, is it possible to track or repeatedly image the same intestinal region over time (using the tattoos)? The axis title of supplementary figure 2 is redundant: "...visible crypts visible".

We thank the reviewer for their attention to detail and questions about ambiguity in our work. Z-stacks of the crypts were taken, but single frames were utilized rather than maximum projections as it is challenging to compensate for respiration motion in these stacks while progressing through multiple levels, and there is some overlap of labeled crypts in different planes. We have clarified this in the methods. Additionally, **Fig. 3** has been amended to show locations which were tracked over multiple days with the help of either the tattoo (**Fig. 3c**) or a vascular dye (**Fig. 3d**).

“We quantified the number of crypts labeled with GFP per time point averaged over 3 regions of colon captured in single planes” (p.15, § Irradiation).

15. The authors describe the imaging of the colonic ENS using Wnt1-cre:tdTomato mice in the results but the purpose of this data in the context of testing the effect of SNS on colonic neuronal firing is unclear. Would it not be more informative to show the GCaMP3 signal in the ENS over time?

We have amended the discussion and methods to improve clarity in our methods and justification. We use Wnt1-cre:tdTomato mice initially because they have stable, high intensity endogenous fluorescence. This first experiment was to demonstrate that we could image the ENS reliably without the complication of a signal with variable strength. We also used this high signal system to establish a protocol to minimize motion artifacts during imaging and processing.

“First, we visualized the myenteric plexus in Wnt1- tdTomato mice over several hours through the colonic window to test our ability to visualize this population while minimizing motion artifact (**Supp. Fig. 5**). We also refined an image processing pipeline to track individual neurons to account for slight movements to further enhance quantification (**Supp. Video 3-6**)” (p. 7, § Sacral innervation and modulation in the colon).

16. The data presented in Fig 4f illustrating Ca²⁺ activity in the colonic myenteric neurons following SNS is only shown for the different frequencies tested, but lacks a comparison with baseline activity without stimulation over a 2 min recording.

In response to the many insightful comments on **Fig. 4**, we have made extensive changes to make it more robust and easier to understand. We compare the effect of each stimulation condition (7, 14, and 21 Hz) to unstimulated control recordings (baseline recordings over a 2-min period). We make these comparisons for each stimulation condition (**Fig. 4c**) and we collapse across stimulation frequency to compare any stimulation condition to the unstimulated condition (**Fig. 4d**).

The data in **Fig. 4b,f** also include a 30-second baseline period before stimulation onset to establish a baseline for each cell.

17. Given the effectiveness of fast blue labelling of the sacral innervation of the colon, is it possible to examine the Ca²⁺ responses of FB-labelled enteric neurons to SNS? This may provide stronger support for the SNS-evoked activation enteric neurons.

We thank the reviewers for this suggestion. We repeated these experiments in four mice and include the results in **Fig. 4e-h** and **Supp. Fig. 7**. We found Fast Blue labeling in 13.5% of GCaMP-positive cells (n = 4). Of the GCaMP-positive cells that were labeled with Fast Blue, 10% responded to SNS (n = 4). Due to relatively low labeling percentage, these experiments are insufficiently powered to make meaningful claims, however they provide intriguing insight into the possible mechanisms of SNS.

This information can be found in the text on **p. 8**, § Sacral innervation and modulation in the colon:

We also traced the sacral nerves to compare their innervation of the lower gastrointestinal tract with the firing responses of colonic myenteric neurons. After injecting Fast Blue, a retrograde dye⁵¹, into the S2 intervertebral foramen, we identified Fast Blue in the colonic myenteric plexus by imaging through the colonic window (**Fig. 4e**; **Supp. Fig. 7**). Intravital imaging through the colonic window revealed Fast Blue-labeling in GCaMP-expressing myenteric neurons, which permitted us to monitor their calcium activity in real-time (**Fig. 4f**). On average, we observed Fast Blue labeling in 13.5% of GCaMP-expressing myenteric neurons in 4 mice (**Fig. 4g**). Of the cells labeled with Fast Blue, 9.8% on average responded to SNS in 4 mice (**Fig. 4h**). These data demonstrate the colonic window can be used to monitor enteric neural activity in real time under chronic, survival conditions.

18. Animals were imaged for several days in a row. Some up to 11 days? With neuron turnover rate as suggested by Kulkarni et al. (cited as reference 74), the authors should have seen around a 60% renewal of neurons. Was this indeed the case?

While the window could be used to study this topic, it was not evaluated in this manuscript due to limitations in the transgenic strains we utilized in our ENS studies. We cannot confidently make any claims about neuronal turnover using these lines. Wnt1-tdTomato mice express fluorescence in all neural crest derived cells, including enteric glial cells. Changes in Wnt1-positive populations in these animals could be due to changes in multiple cell types, and not strictly neurons. Pirt-GCaMP3 mice do not express GCaMP3 in all myenteric neurons, only a subset that are Pirt-positive. Further, GCaMP is a dynamic fluorophore driven by intracellular calcium, so it would be difficult to determine enteric neuron turnover in this transgenic line.

19. A DSS model should not be called an IBD model. DSS causes colitis indeed, but it does not necessarily mimic IBD.

We have revised our language in the manuscript. “Next, we imaged immune cell activation in a live, chronic murine colitis model to visualize immune response in the colon” (**p. 6** § Live monitoring of immune response to an inflammatory challenge in the colon *in vivo*).

20. Last paragraph of the introduction contains some repetition (epithelium, stem cell, microbiota...) that is best avoided.

We have revised our language in the manuscript. “Using this technology, we demonstrate real time imaging of colonic stem cells, immune cells, bacteria, and luminal content passage in live animals. Furthermore, using mice with transgenic calcium indicators, we observed spatiotemporal neuronal activity in the colon in

response to SNS, showing direct evidence that SNS activates enteric neurons *in vivo*. This easy-to-use colonic implant technology will be a useful tool in furthering our understanding of colonic diseases” (p. 4, § Introduction).

21. Details about the NY staining and FB imaging are not provided. Was FB imaged using two photon excitation?

We have removed the content which included Nuclear Yellow staining, which was only used in conjunction with post-mortem immunostaining.

We revised the methods to reflect the *in vivo* imaging of FB using two-photon excitation. “Fast Blue was visualized *in vivo* with multiphoton imaging” (p. 16, § Nerve Tracing).

22. The quantification of fast blue labelled enteric neurons across the different gut regions apparently lacks error bars in Fig.3. Was more than 1 mouse examined?

Only one animal was examined, and this was not performed *in vivo*. We have removed this experiment from the revised manuscript to provide more rigorous data and exclude studies that are not pertinent to the colonic intravital microscopy.

23. What do the authors mean by “900 Hz stimulation” when referring to multiphoton imaging?

This was a typo and intended to be 900 nm excitation wavelength. This section was removed when we removed the *ex vivo* calcium imaging methods section.

We have ensured the other excitation/emission wavelengths are reported in the appropriate units.

“Second harmonic generation was performed with an excitation wavelength of 850 nm and a detection gate between 400-450 nm” (p. 13, § *E. coli* and Fluorescent Dye Gavage).

24. What is meant with “voltage sensors calcium indicators” in the last paragraph of the discussion?

We have revised our language in the manuscript for clarity.

“Physical stabilization via the ferromagnetic scaffold is supported by existing tools for image post-processing to track single cells, similar to methods used in the brain, spine and heart⁵⁵. The combination of our colonic window design and ferromagnetic scaffold overcome the constraints of colon physiology and can be further expanded to other moving tissues and mouse models” (p. 9, § Discussion).

25. The description of the Ex vivo tissue imaging methods reads very abruptly in certain places: “Tissues in bicarbonate buffer and put over a 10 FR Foley catheter” and “Cathode was connected to steel clamp and anode placed in solution”.

We appreciate the helpful feedback. This section was removed from the manuscript when we removed the *ex vivo* calcium imaging methods.

26. The paper should be carefully proofread for typographical errors and syntax. E.g. Methods: ... mice... were treated with eye ointment; SupFig 4: colon shown stained cells for...

We thank the reviewer for their detailed notes on our work. We have made substantial revisions to the manuscript with particular attention to typography and syntax.

Reviewer #2

1. The only point the author may clarify "the custom-designed implant was inserted under the colon and sutured to the skin" - do the thread run from the borders of the colon to the skin at the border of the window ?!

We revised the manuscript to clarify the scaffold location and implant process.

From the Methods and Materials, **p. 11,12**, § Window Design and Implantation:

All implants and surgical equipment were sterilized in an autoclave before use. Surgery was conducted on adult mice using sterile technique, isoflurane anesthesia, and analgesics as described previously¹¹. The ferromagnetic scaffold was held in place by loops of suture passed through the anchor points and sutured to the abdominal wall on either side of the window, providing 2-3 mm of slack so the scaffold could be displaced during colonic distension or animal motion. The surgical site was marked on the surface of the colon intraoperatively with a tattoo using the AIMS tattoo system (Braintree Scientific, ATS-3 Q). The success rate of reversible, consecutive, magnetic displacements was 100%, and mice survived past two weeks without any adverse events before reaching the experiment endpoint.

REVIEWERS' COMMENTS:

Reviewer #1 (Remarks to the Author):

The paper has improved substantially. However there are still a couple of issues that need some attention.

The acclaimed second harmonic signal in figure 2 (top panels) looks much more like autofluorescence that is generated by the crypts. Are the authors sure that this is not the case?

The authors state in the legend of Figure 3a that the same position was tracked for 5 days and alludes to use of the fluorescently-labelled vasculature for this purpose. However, the text in the Results section explains the loss of the Texas red-dextran signal by day 3. Hence, the figure which is suggestive of the finding that activated monocytes are observed in the same location imaged up to 3 or even 5 days is misleading, unless the tattoo was used for tracking (in which case this is unclear).

In the results, the colon of the same irradiated Lgr5-eGFP mice were described to have been tracked over a week (line 129-131) but data are only shown for up to day 3 post-treatment (Figures 3b, c).

As an aside remark, why does the photo of the window at day 13 looks that cloudy? Is this always the case 2 weeks after surgery?

1. The acclaimed second harmonic signal in figure 2 (top panels) looks much more like autofluorescence that is generated by the crypts. Are the authors sure that this is not the case?

We thank the reviewer for their detailed comments on our revised figures. In response to the original concerns regarding possible detection of autofluorescence rather than second harmonic generation, we performed an additional experiment whose results are in **Supplementary Figure 3** showing that we did not detect significant autofluorescence in the broad collection gate which did not include the 50 nm detection window centered around the second harmonic wavelength of 425 nm. We have amended the text to draw attention to this additional validation:

“Colonic crypts were imaged using second harmonic generation microscopy. We confirmed the microscopy techniques have low autofluorescence by imaging through the colonic window in C57BL6/J mice (Supp. Fig. 3). While second harmonic generation is possible in the non-transgenic mice, excitation with the utilized wavelength of 850 nm does not produce noticeable autofluorescence in the capture gate above 450 nm.” (p. 5, § Imaging of bacteria and luminal content in the colon *in vivo*).

2. The authors state in the legend of Figure 3a that the same position was tracked for 5 days and alludes to use of the fluorescently-labelled vasculature for this purpose. However, the text in the Results section explains the loss of the Texas red-dextran signal by day 3. Hence, the figure which is suggestive of the finding that activated monocytes are observed in the same location imaged up to 3 or even 5 days is misleading, unless the tattoo was used for tracking (in which case this is unclear).

We have clarified the text to indicate the tattoo was used for tracking. We have also amended the legend of Figure 3 accordingly:

“Figure 3. In vivo imaging of the same location over time via the colonic window. (a) Representative images from a CX3CR1GFP x CCR2RFP mouse before and after DSS treatment (administered on Day 0). Using a combination of tattoo and vascular mapping we track innate dendritic cells and inactivated monocytes (green), activated monocytes (blue), and vasculature (red) in the same position for 5 days.”

3. In the results, the colon of the same irradiated. Lgr5-eGFP mice were described to have been tracked over a week (line 129-131) but data are only shown for up to day 3 post-treatment (Figures 3b, c).

We thank the reviewer for catching this typo leftover from the previous data. We have amended the text to accurately reflect the study presented in Figures 3b-c:

“To demonstrate that our intravital colonic imaging can capture the dynamics of these cells, we treated Lgr5-eGFP17 mice with 18 Gy irradiation and tracked the

colon over five days in the same mice.” (p. 6, § Tracking colon epithelial stem cell populations *in vivo*).

4. As an aside remark, why does the photo of the window at day 13 looks that cloudy? Is this always the case 2 weeks after surgery?

We often observe some mucus build-up on the interior of the window, which can contribute to a cloudy appearance in photographs.